# EMERGENCE OF SURPRISE AND PREDICTIVE SIGNALS FROM LOCAL CONTRASTIVE LEARNING

## ABSTRACT

Hierarchical predictive models are a popular view of cortical representations, and may hold promise for enhancing the robustness and generalization of machine learning architectures. These models exploit the local computation of predictive signals to pass information, but their manifestation in neurobiology remains rightly debated. This paper contributes an emerging principled approach to this discussion by investigating the inverted Forward-Forward Algorithm, a biologically plausible approach to learning with only forward passes. We demonstrate that hierarchical predictive signatures can emerge from a parsimonious combination of contrastive learning and learned cancellation which shape the network's representation across layers. We identify similarities between our model and hierarchical predictive coding, and characterize the emergent properties of the resulting representations. These properties of the inverted FF model present falsifiable predictions which may be accessible in emerging experiments. This work advances the hypothesis that the computational properties which emerge in neocortical circuits, widely acknowledged as the basis of human intelligence, may be attributed to parsimonious, local learning principles.

## 1 INTRODUCTION

The neocortex contains hierarchically layered circuits with rich feedforward and feedback connections (Chaudhuri et al., 2015; Bassett & Sporns, 2017; Siegle et al., 2021). The feedforward (or bottom-up) pathway involves the transfer of information from lower-level sensory areas to higher-level association areas, leading to the extraction of input-specific features. In contrast, the feedback (or top-down) pathway aids in the integration of high-level information by relaying signals from higher-level areas to lower-level ones. Though often assumed to propagate learning-related errors (LeCun et al., 2015), the functional role of feedback connections has been implicated in many different perceptual and cognitive abilities such as attention, efference copies, memory retrieval, etc. (Mechelli et al., 2004; Gilbert & Li, 2013).

In the Bayesian view of cortical feedback, the bidirectional flow of information enables integration of ongoing sensory inputs with existing cortical representation of prior contextual information (Badre & Nee, 2018; Khan & Hofer, 2018; Froudarakis et al., 2019). A specific way to convey such contextual information is through surprise- and familiarity-based signals. When incoming sensory input corresponds to expectations, the surprise signal is minimal. When the input deviates from expectations, the surprise signal rises, indicating novelty or unfamiliarity. It has been shown that novel stimuli elicit increased neural activities that decrease over repeated presentations (Garrett et al., 2023; Piet et al., 2023). Such surprise signals contain information for the brain to improve its internal model of the external world, which leads to more refined and accurate expectations that can direct behavior Wolpert et al. (1998); Kawato (1999); Schenck (2008).

However, the mechanisms through which feedforward and feedback connections interact and generate such surprise signals are not well understood. An influential theory in neuroscience, predictive coding (Rao & Ballard, 1999; Jiang & Rao, 2022), postulates that feedback circuits deliver top-down spatiotemporal predictions of lower-level neural activities, while feedforward circuits send bottom-up prediction errors (surprises) to higher levels. Despite its popularity, minimizing prediction error is an iterative process based on gradient descent, which requires physical time for convergence and implies the symmetry between the feedback and feedforward synaptic weights, limiting its biologi-

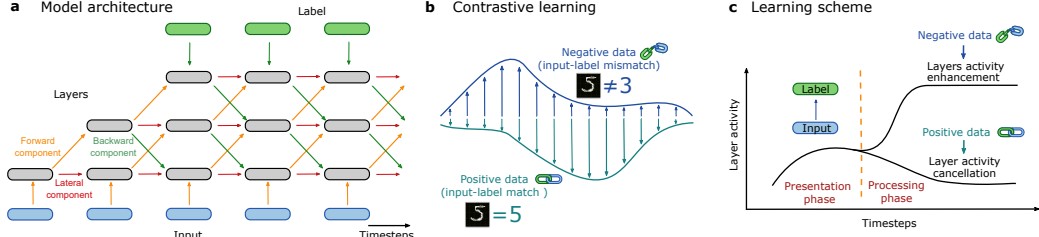

Figure 1: Simple illustrations representing model architecture and learning scheme. a) Model architecture is shown where data inputs are clamped to the bottom and label inputs are clamped to the top of the network. b) Forward-Forward contrastive learning schematic with definition of positive and negative datasets, where the label mismatches or matches the sensory input. c) Learning scheme of the model is shown where the training phase proceeds in two steps - the presentation and processing phase respectively. In the processing phase, positive data should have a low activity, whereas negative data should have a high activity.

cal plausibility (Rao & Ballard, 1999; Lillicrap et al., 2016). Additionally, computing the prediction error requires a one-to-one correspondence between the predictive neurons and error neurons, which have not been confirmed experimentally (Jordan & Keller, 2020).

Here, we present a simple and biologically plausible mechanism that captures the spatiotemporal predictive nature of cortical processing without generating explicit predictions. Our model is based on the Forward-Forward model (Hinton, 2022), a recently introduced form of contrastive learning. We inverted the original Forward-Forward objective to minimize the activity of positive training data and maximize the activity of negative training data, where we now refer to the level of activity as surprise. Such an objective promotes activity cancellation when top-down labels match bottom-up sensory input. As a consequence, different layers across the hierarchy learn to predict each other's activity to enable such minimization (or cancellation) of activities.

Our most significant contributions are:

- we demonstrate that our model reproduces both hierarchical and temporal properties of predictive computations by means of generating information flows that lead to surprise and cancellation signals (Secs. 3.1 to 3.2);

- we illustrate a mechanistic understanding of the emergence of such information fluxes by tracing their origin to the circuit's ability to implement spatiotemporal cancellation across layers of activity (Secs. 3.2 to 3.3)

- we establish an equivalence between our contrastive learning rule and a distinctive three-factor Hebbian plasticity, showcasing strong connections (and differences) to predictive coding. This finding emphasizing the biological plausibility of our model, characterized by online capability, no-weight transport, and the integration of local signals with a global signal. (Sec. 3.5).

These results demonstrate that the application of a fundamental contrastive learning technique that integrates surprise and cancellation dynamics generates predictive spatiotemporal properties. This suggests that these properties, which are generally regarded to be distinctive characteristics of neo-cortical computations, can be generated by simple, locally-defined learning principles.

## 2 MODEL ARCHITECTURE AND LEARNING SCHEME

We extend the Forward-Forward model (Hinton, 2022; Ororbia & Mali, 2023; Ororbia, 2023), a back-propagation-free learning paradigm regarded as a form of contrastive learning. Our model consists of a hierarchical network with multiple layers, with label information clamped at the top and input at the bottom (Fig. (1a)). The label in this architecture is thought of as a second input, where the output of this network is it's magnitude of activity. The activity magnitude serves as an indication of how well the label input (clamped top) matches the supplied data input (clamped bottom).

Additionally, the network operates in the time domain, where activities of each layer are updated based on activities of adjacent layers at the previous timestep. The information flow of the input is considered to be bottom-up, whereas the information flow of the label is considered to be top-down. Neurons in each layer receive presynaptic input from a lower layer, a higher layer, and itself at the previous timestep. For layers at the bottom and top of the network, where there is no layer below or above, the presynaptic input is taken from the data input or label input respectively.

The training process involves defining positive and negative datasets. Positive (negative) data is one in which the label and input do (not) match. For the negative dataset, the activity of each layer is increased, which can be thought of as indication of surprise (Fig. (1b)). For the positive dataset, the activity is diminished resulting in decreased surprise.

In this network, each layer acts as its own learning agent based on its 3 inputs (top-down, bottom-up, lateral), to produce a level of activity representative of whether the data input matches the label input, even if the data and label inputs are not directly adjacent to the layer itself. The learning objective is defined in terms of the individual layer activations at each time $\vec{x}_{\text{layer}}(t')$:

$$\mathcal{L}_{\text{layer}} = (-1)^{\eta}\sigma(\vec{x}_{\text{layer}}^{T}(t')\vec{x}_{\text{layer}}(t') - \theta) \tag{1}$$

where the parameter $\theta \in \mathbb{R}$ is the threshold set for the surprise calculation, while $\eta$ equals zero or one ($\eta = 0, 1$) for negative and positive data samples respectively. The final ingredient is the non-linearity $\sigma$ which is taken to be a soft-plus function.

It is worth noting that, while global supervisory terms are often dismissed as biologically implausible, $\eta$ here acts as a simple, singular global signal, rather than something more complicated. Neurotransmitters in biological networks can act over a wide area (via volume transmission) to modulate the activity of many neurons, rather than just those directly connected by synapses. As such, a type of non-local signal might occur in response to learning or other processes, and remain biologically plausible. (We expand on bio-plausibility in section 4.3.)

Training involves conducting forward passes for positive and negative data simultaneously. The most essential aspect of training is that only forward passes are made at all times. This makes the Forward-Forward algorithm biologically plausible, as it avoids the non-locality of weight information inherent to backpropagation. Only layer-local weights (top-down, bottom-up, lateral) are modified during training. In addition, we demonstrate that this algorithm is theoretically equivalent to particular types of Hebbian learning (see Supplementary Material). The network learns to process input and label by combining bottom-up and top-down information flows to generate activations that reflect positive and negative data points, respectively, upon training.

We revise the training procedure to increase similarity to signal processing in biological networks (Fig. (1c)). First, the input is presented to the network (presentation phase), followed by the generation and introduction of the label (processing phase). Each phase is comprised of a number of timesteps (10 and 15, respectively). Training does not occur during the presentation phase; rather, training occurs only when label information is received during the processing phase. Each layer's surprise decreases (increases) based on whether the associated label matches (mismatches) the presented input. This resembles cortical processing where surprise signals, often resulting in increased activity, are thought to arise from a mismatch between our sensory inputs and top-down signals reflecting our internal beliefs or world model – here, the label information. In other words, when the input and label match each other, the activity is lower than when a mismatch, or surprise, occurs.

Inference involves running both the presentation phase and the processing phase, but keeping all parameters fixed. The class is then able to be decoded off of the latents of the processing phase.

## 3 RESULTS

We train the model on the MNIST dataset following the scheme highlighted in Fig. (1). For every iteration, a single MNIST image is selected and presented as an input to the network (presentation phase of 10 timesteps). Following this presentation phase the label is introduced, while still presenting the image, and the network processes both input and label information (processing phase of 15 timesteps). We first focus on the spatial integration of bottom-up and top-down information flows. Learning follows as per Eq. (1) and accuracy is computed as outlined in Hinton (2022): for each

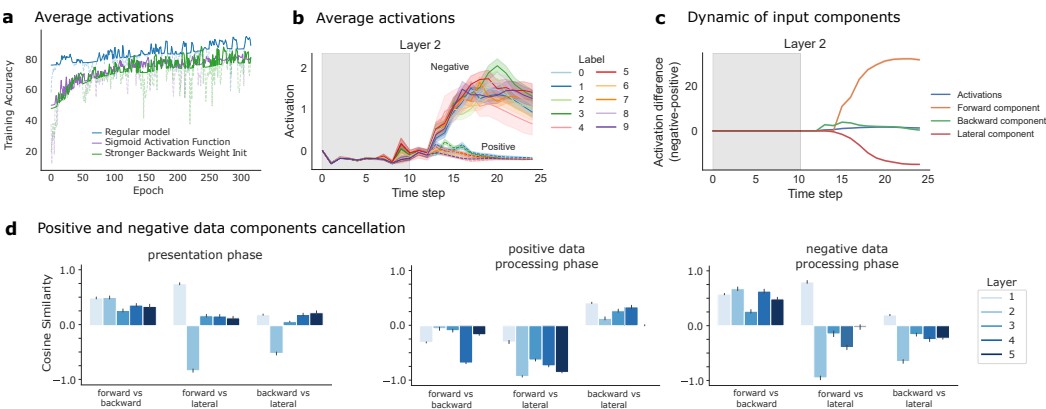

Figure 2: Figures for validation accuracy, layer-wise activation progression throughout time, layer-specific cancellation, and cancellation patterns between the components forming a layer's activation update. a) Accuracy (y-axis) over time (x-axis) is shown for various configurations of the network. We show a deterioration for both a stronger label clamped weight initialization, as well as with a sigmoid activation instead of leaky ReLU. b) For the second layer in the network, the average activations (y-axis) obtained over 1000 samples are shown per class over time (x-axis), for both positive and negative data. Negative data induces a large and sustained surprise signal with raising activities after the label is shown. Positive data has a very small surprise and returns to baseline low activity. c) For the second layer in the network, the activations for positive data averaged across 1000 samples, are broken into their pre-synaptic components (y-axis) and plotted across time (x-axis), which show strong cancellation as indicated by the resultant summed post-synaptic activity (blue). d) Cosine similarity (y-axis) of positive and negative data activations for the presentation and processing phase (before and after label presentation). The alignment patterns across components highlight the different cancellation profile at work during the processing of positive versus negative data.

input image $x$ all possible labels (classes 0 to 9) are introduced to the network, we deem the input image to be accurately processed if the surprise for the correct label is lower than for any other label.

We train a 5 layer network with 700 neurons per layer minimizing Eq. (1). We used RMSProp as the optimizer with learning rate $5 \cdot 10^{-5}$, batch size 500, and Leaky ReLU as the transfer function for all units. We use no momentum and applied a stopgrad operation to all adjacent layer activations to prevent the parameter gradients from growing beyond one-step. Weight initializations and further details can be found in the available repository [1]. The 5-layer, 25 timestep model accuracy achieved 95% test accuracy upon training (Fig. (2a)). Different activation functions were attempted, and sigmoid consistently showed to perform worse than ReLU derivatives, Fig. (2a).

## 3.1 HIERARCHICAL EMERGENCE OF SURPRISE AND CANCELLATION SIGNAL

By analyzing layer activity via L2 norm over time, we were able to confirm that the model learned to dynamically suppress neural activity across both layers and time whenever the input image matched the respective label (Fig. (2b)). The difference between negative and positive activations showed a clear divergence upon label presentation, Fig. (2b). This trend was the result of the contribution of multiple input components to the layers, Fig. (2c). Notably, the forward component representing the input from lower layers was the only significantly stronger component for negative versus positive data, suggesting a leading role of this component in driving the increased activity for surprise signals. This was true across all layers (Fig.S2). In order to understand how these input components were driving the increase in activity for negative data (surprise signal) – and decrease in activity for positive data (the cancellation upon label presentation), we focused on the late timesteps (10-25) where such phenomena appeared. We verified whether different input components were aligned or misaligned with each other, therefore issuing a cancellation in the overall activities. To this end we computed the cosine similarity (scalar product) between all pairs of the three input components before and after label presentation (presentation vs processing phase Fig. (2d)). For positive data the forward component was largely anti-aligned to both the backward and lateral components, suggest-

---
[1] github [REDACTED FOR REVIEW]

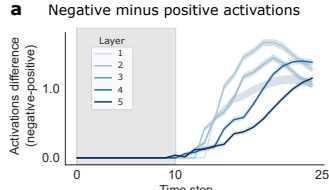 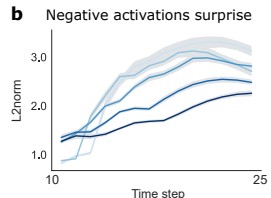 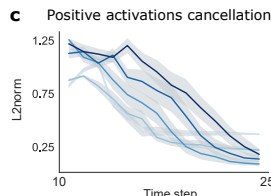

Figure 3: Activations surprise and cancellation order. a) The negative minus positive activations (differences) over time (x-axis) are shown as a measure of the negative activation surprise signal, offset from the baseline of our positive actions. b) The L2 norm of negative activations (y-axis) is shown across time (x-axis), visualizing the cancellation cascade during the processing phase. c) Same as panel b for the cancellation of positive activations.

ing that the decrease in activity was due to the bottom-up (forward) information flow canceling the top-down (backward) and recurrent (lateral) information flows (Fig. (2d)). Conversely, for negative data, the top-down and bottom-up information flows showed a higher degree of alignment, resulting in increased activations (surprise signal) (Fig. (2d)).

This analysis shows that our model reproduces hierarchical properties of predictive computations by generating information flows that result in surprise and cancellation signals. These signals are associated with the processing of negative and positive data, respectively, and involve distinct network information flows based on the dynamic cancellation of multiple input components. Although the degree of alignment across components could vary from instantiation to instantiation the these cancellation phenomena were highly robust.

## 3.2 DYNAMICAL EMERGENCE OF SURPRISE AND CANCELLATION SIGNALS

We next interrogated the temporal characteristics of the cancellation and surprise information flows. We began by plotting activation differences across all layers (as performed in Fig. (2c)) in Fig. (3a). This demonstrated that the encoding of positive versus negative data diverged more rapidly between early layers compared to later layers. To confirm this, we analyzed activations for negative and positive data during the processing phase, after introducing the label. For negative data activity grew faster for earlier layers despite label information being fed from the top of the hierarchy Fig. (3b). In the case of positive data early layer activations led the cancellation cascade by returning to a lower activation state, prior to late layers establishing a bottom-up cancellation ordering. We also analyzed the cosine similarity between activations of consecutive layers for both positive and negative data (Fig.S3) confirming these cascade ordering respectively for surprise and cancellation signals. Together, these findings shown in Figs. (3a) to (3c) indicate that alignment and anti-alignment dynamics across layer activations, leading to surprise and cancellation signals, originate in early layers despite the introduction of the label at the top of the hierarchy.

## 3.3 INTERPRETING LATENT REPRESENTATIONS WHICH DRIVE CANCELLATIONS

In order to understand the latent space mechanics driving cancellations on positive data, we sought to understand the intricate mechanics governing the latent space dynamics. We first plotted the average class-wise activations for various PCs in lower dimensions (Fig. (4a)). We observe that the lower-order PCs do not offer a strong representation of the class, but they do offer a consistent path through the space that starts and ends at the same point. This is in line with the mechanics of the network under positive data, which starts from an initially low activity and recovers to a similarly low activity following all the timesteps where the label is presented. In the higher-order PCs (4-6), chosen for their stronger representations, the same looping mechanics are shown. However now the classes are represented in a separable manner. To further quantify this qualitative analysis we performed a careful decoding analysis highlighting the presence of label information across multiple PCs (Fig.S4c). For negative data, the looping behavior in the latent space does not occur, and the latent states drive away from the origin erratically in a class and label dependent manner.

To measure the directionality of information flow throughout, 5 MLPs were trained on the latents of each of the 5 layer-wise activations for positive data (Fig. (4b)). High decodability indicates

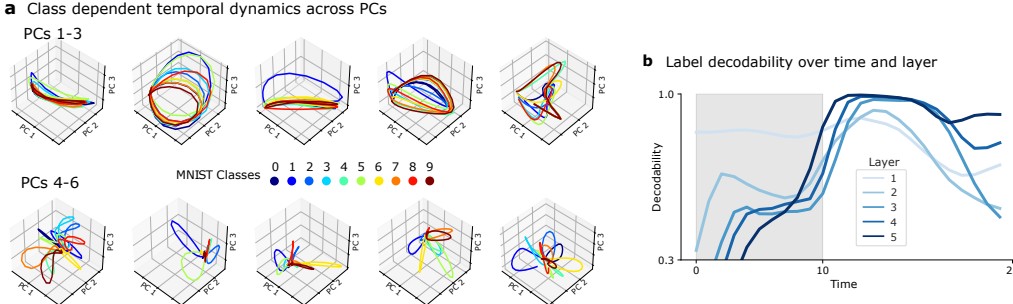

Figure 4: Latent representations and label decodability over both principal components and layer. a) Representation of the layer-wise latent spaces on three dimensions via PCA where classes are represented by color. The first three PCs are shown to indicate a lack of class separability. Higher-order PCs are shown to indicate stronger class separability. b) Decodability (y-axis) over time (x-axis) for different layers, indicating that the pre-label timesteps are driven by a distinct bottom-up temporal driving, and by contrast late processing time displays cancellation (and the associated drop in decodability) cascading from the bottom-up.

label specific information. Two distinct cascades of decodability increase were observed: one for the presentation phase (timesteps 0-9, upon image presentation), and another for processing phase (timesteps 10-24, upon label introduction). These decodability increases revealed a layer-wise temporal ordering in opposite directions, consistent with the introduction of the image and label respectively at the top and bottom of the layer hierarchy. In the presentation phase, the decodability increases first for lower layers, indicating a bottom-up temporal ordering. By contrast, in the processing phase, the decodability increases first for upper layers. This analysis examines population-level coding and demonstrates the encoding properties of the network throughout its hierarchy tracking label specific information.

### 3.4 COMPARISON OF THE FORWARD-FORWARD ARCHITECTURE WITH PREDICTIVE CODING NETWORKS

The hierarchical predictive dynamics analyzed this far, giving rise to surprise and cancellation signals, are specific of our model. We compare our model with established predictive coding networks (PCNs), first introduced by Rao & Ballard (Rao & Ballard, 1999), to further characterize these dynamics in contrast to those of PCNs. Predictive coding networks are characterized by a hierarchical structure wherein each layer predicts the subsequent layer's activity, informed by the product of its activity and a weight matrix, processed through a nonlinear function. The objective function of this predictive coding network is to minimize the loss:

$$\mathcal{L}_{\text{layer}}(t) = |\phi(B\vec{x}_{l+1}) - \vec{x}_l|^2 \,, \tag{2}$$

which is often referred to as prediction error. Here, $x_l$ denotes the activity of layer $l$, $B$ represents the weight matrix, and $\phi$ is a nonlinear activation function. The training process involves an alternating optimization strategy where the network first adjusts its weights to minimize the prediction error and subsequently refines the layer activations to further reduce the discrepancy between prediction and actual sensory input. This iterative process aims to model the brain's learning mechanism, which continually adapts to new information.

The Forward-Forward architecture differs from this framework in its intrinsic generation of predictions. Rather than relying on a hand-coded error computation between layers with dedicated error neurons and prediction errors, the Forward-Forward network learns predictions through a local learning rule intrinsic to each layer. This critical difference generates a dynamic which is qualitatively different. Figs. (5a) to (5c) show respectively the activations of error neurons, non-error neurons, and the compound activity. None of the highlighted phenomena in the Forward-Forward dynamics is present in such a predictive coding model. Critically, in a PCN, there is no distinction between positive or negative data, no surprise or cancellation signals generated by the network, and there is no bottom-up (or top-down) cascade in the way information propagates through the networks. On the other hand, these elements are observed in cortical networks, and are naturally generated by the Forward-Forward model.

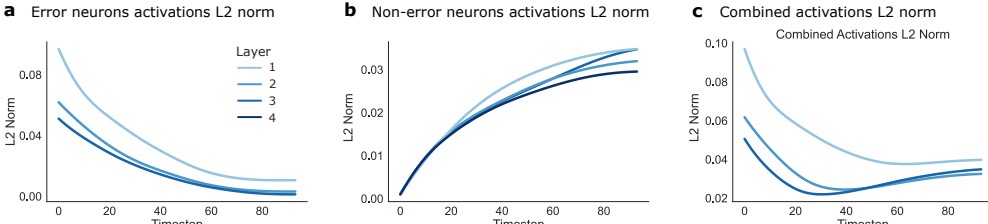

Figure 5: Analysis of Predictive Coding Network a) The norm of activation of error neurons decreases with time for all layers. b) Norm of activation of non-error neurons increases across timesteps with no relationship between layer amplitude layer position.c) Norm of compound activations of error and non-error neurons.

The Forward-Forward's approach of eschewing hand-coded prediction errors leads to more biologically aligned phenomenona, as it appears to reproduce the spatio-temporal bottom-up activity cascade observed in mice full field flash experiments, as highlighted by Siegle et al. (2021). This cascade did not appear in our implementation of a PCN, which instead demonstrated a rise in prediction errors across all layers simultaneously. Further research is needed towards understanding the conditions under which predictive coding networks (PCNs) might align with the Forward-Forward architecture's distinctive dynamics, challenging the boundaries of these computational models in cortical computation emulation.

### 3.5  THE INVERTED FORWARD-FORWARD IS A CONTRASTIVE THREE-FACTOR-LEARNING RULE WHICH CONVERGES TO SYNAPTIC DRIVE CANCELLATION

In this section, we demonstrate an online learning variant, showing learned cancellation arising from minimizing/maximization cumulative surprise.

For a $N > 3$ Forward-Forward architecture where $N$ is the total number of layers, $I$ is the data input, and $\ell$ is the label input, the dynamics of each layer are governed by

$$\dot{\vec{x_1}} = \phi\left(W_1\hat{x}_1 + F_1I(t') + B_1\hat{x}_2\right)$$
$$\dot{\vec{x_i}} = \phi\left(W_i\hat{x}_i + F_i\hat{x}_{i-1} + B_i\hat{x}_{i+1}\right)$$
$$\dot{\vec{x_N}} = \phi\left(W_N\hat{x}_N + F_N\hat{x}_{N-1} + B_N\ell(t')\right).$$

The locally defined loss for a one-step update takes the form of

$$\mathcal{L}_{layer}(t') = (-1)^\eta \sigma(\vec{x}_i^T \vec{x}_i - \theta)$$

where $\sigma(x)$ represents the softplus as $\sigma(x) = \log(1 + e^x)$ and is a smooth version of the ReLU nonlinearity. Importantly, the dynamics of the $\eta(t')$ are governed by a bistable dynamics:

$$\eta(t') = \delta_{L(t'),I(t')}$$

which occurs with long timescales between switches. Here $\delta_{ij}$ is the Kronecker delta notation, and $L(t')$ a signal that defines positive and negative data. This phenomenon could align with neuromodulator-induced shifts in underlying dynamics. It also suggests a criterion for selecting $\eta(t')$ based on the instantaneous surprise of the stimulus against the speculative label. Although beyond the scope of this work, the closure of this loop between activations and cost function may generate valuable insights into unsupervised variants of these learning rules (Ororbia & Mali, 2023).

We then execute a single step-gradient update in each parameter:

$$\partial_t W_i = -\alpha\nabla_{W_i}\mathcal{L}(t')$$
$$\partial_t B_i = -\alpha\nabla_{B_i}\mathcal{L}(t')$$
$$\partial_t F_i = -\alpha\nabla_{F_i}\mathcal{L}(t').$$

By iteratively minimizing this locally defined objective function, we seek a hierarchical structure that will work in concert with the other layers to minimize activations for positive data. However, when there is a data mismatch between label and image class, the representations will learn to avoid cancellation to increase their surprise.

Indeed this single-step update for a given layer takes the form of a three-factor Hebbian learning, since

$$\nabla_{W_i}\mathcal{L}_i(t') = (-1)^\eta \underbrace{\sigma'(\vec{x}_i^T(t')\vec{x}_i(t') - T)(\vec{x}_i(t'))}_{\text{Gating factor}} \underbrace{\phi'(z(t'-1))}_{\text{Pre-synaptic current}} \underbrace{\vec{x}_i(t'-1)}_{\text{Post-synaptic firing rate}},$$

where $\vec{z}(t'-1) = W_i\hat{x}_i(t'-1) + F_i\hat{x}_{i-1}(t'-1) + B_i\hat{x}_{i+1}(t'-1)$ is the input current into the nonlinearity.

This form of learning is formally a gated Hebbian or three-factor rule (Bahroun et al.; Kuśmierz et al., 2017; Bredenberg et al., 2021; Pogodin & Latham, 2020; Portes et al., 2022; Bellec et al., 2020; Murray, 2019) linking the locally defined objective function to the product of the pre-synaptic current and the post-synaptic activation. Crucially, this loss for a single trial is indeed the cumulative sum over the full sequence:

$$\Delta x_i = \alpha \sum_{t'} \nabla_{x_i}\mathcal{L}_i(t').$$

These gradients have important implications on the shape of the learned solutions. These learned solutions (where the gradient goes toward zero can occur under a number of conditions). These conditions include the direct cancellation of the synaptic drive currents (input components) governing the time dynamics of the hidden layer: $W_i\hat{x}_i + F_i\hat{x}_{i-1} + B_i\hat{x}_{i+1} = 0$.

In this section, we established an equivalence between the inverted Forward-Forward and a unique form of gated Hebbian plasticity (dominated by a slow timescale sign-flip, a threshold passing gate, and a self-gain) (see section 4.1. In addition, we have demonstrated that the cancellation mechanism of lateral, top-down, and bottom-up signals satisfies the stationary solution to this deduced three-factor learning rule.

# 4 DISCUSSION

## 4.1 WHY IS THE PCN DISTINCT FROM THE INVERTED FF?

On the surface, the PCN and inverted FF are motivated by the same ambitions. They both avoid back-propogation in favor of local cost functions that admit three-factor descriptions of their learning. By using this unifying approach to focus on the the third factor alone, we can appreciate the differences more clearly.

$$\text{Third Factor} = \begin{cases} (-1)^\eta \sigma'(\vec{x}_i^T(t')\vec{x}_i(t') - T)(\vec{x}_i(t')) & \text{for the inverted FF} \\ (\phi(B_i\vec{x}_{i+1}) - \vec{x}_i) & \text{for the PCN} \end{cases} \tag{3}$$

The inverted FF differs from the PCN in two key aspects: contrastive supervision and gating conditions. In the inverted FF, supervision involves clamping the label at the top and employing a contrastive signal. In contrast, even in the PCN with a clamped label, the contrastive signal is absent. The second distinction lies in the conditioning of weight updates—on layer activity in the inverted FF and prediction error in the PCN. These differences account for the variations in steady-state activity and ordering.

## 4.2 MECHANISMS BEHIND CANCELLATION ORDER

As a result of our simulation, a compelling non-trivial logic has emerged from the model's hierarchical predictive dynamics, which are often difficult to comprehend. Here, we seek to explain the fundamental mechanistic principles underlying the network's information flow generation. We determined that despite the fact that such insights are difficult to isolate or prove, they may still be necessary to comprehend the model's inner mechanisms.

For the initial presentation phase of both positive and negative data, the image representation flows from the bottom-up. The differences are, however, quite different in the processing phases. For negative data, the processing phase induces a top down signal carrying label infrmation downward through all layers. This top-down label signal causes relatively small increases in layer activation magnitude. It is only when the label information reaches the bottom, that the activation response grows dramatically, indicating a mismatch and evoking a large and sustained exitatory surprise. For

positive data, the label representation traverses to the bottom layer without inducing cancellations, whereby the cancellations then start in a bottom-up manner, despite the top-down label representation.

As the presentation phase blends with the processing phase, it is insightful to note that, neither for the positive nor negative case is some predisposed behavior taking place. The layers in the network do not amplify or reduce their activations until the label representation reaches at least one layer below them and sends a label-infused representation back upwards. This suggests the bottom-up cancellation could be a result of the network's optimizing drive to alter activities when in a familiar state relative to training. During training, there is a continuous and consistent exposure to label-augmented activations, particularly past the early stages, which ingrains a behavior within the network. The network recognizes label saturated activations as the dominant trend it should ideally be prepared for. Given this recognition, the network is best equipped for cancellation when it encounters activation components (forward, backward, recurrent) with label information coming from all components, not just from the top. Under this concept, a layer lower in the hierarchy would be predisposed to cancel first, due to the lower number of potential layers below lacking label-infused information which would thereby block cancellation. Thus, cancellation in lower layers would be followed by a transmission of label-infused activities upwards to the next layer to induce subsequent phases of bottom-up cancellation.

### 4.3 BIO-PLUASIBILITY OF THE INVERTED FORWARD-FORWARD

The inverted Forward-Forward model uses activation contrast to navigate credit assignment in hierarchical architectures in a bio-plausible fashion by incorporating: the absence of weight transport (Lillicrap et al., 2016; Portes et al., 2022), online compatible learning rules consistent with three-factor Hebbian plasticity, a biologically analogous separation of timescales, and the incorporation of structural hierarchy. First, feedback is separated from backpropagation and instead incorporated as a top-down signal avoiding weight transport. By disconnecting the $F$ and $B$ matrices, the flexible learning rule finds aligned but non-weight transported solutions.

Although global supervisory terms are sometimes downplayed as biologically implausible, it is worth comment that $\eta$ functions as a simple, singular global signal. In biological networks, neurotransmitters can exert influence over a broad area via volume transmission, modulating the activity of many neurons beyond those directly connected by synapses. This non-local signaling introduces biological plausibility, especially in the context of the local update rules of the inverted Forward-Forward model, which involve local Hebbian plasticity gated by thresholded activation and signed by data type. The associated third factor ties an external signal, suggestive of the aforementioned neuromodulatory input, to the minimization (maximization) of layer activity for positive (negative) labels. The slow timescales of this switching, relative to both dynamics and plasticity, suggest a normative hypothesis for the role of perhaps overlooked small molecules (Kuśmierz et al., 2017). Thus, the inverted Forward-Forward model introduces biological plausibility to the direct competition of top-down and bottom-up signal processing, with intriguing implications for interpreting the hierarchy of biological systems (Siegle et al., 2021; Garrett et al., 2023).

## 5 CONCLUSION

In this work, we have presented a biologically plausible mechanism that sheds light on the spatiotemporal and predictive nature of cortical processing without necessitating explicit predictions. Drawing inspiration from the Forward-Forward model, an emerging form of local, contrastive learning, we inverted its original objective function to reduce surprise activations for positive data. This inversion incentivizes activity cancellation between information flows when top-down labels align with bottom-up sensory input. As a consequence, layers across the hierarchy develop the ability to predict and cancel each other's activities, facilitating the minimization of layer surprise.

In conclusion, our study demonstrates a local contrastive learning approach based upon surprise can recreate predictive spatiotemporal properties. The emergence of these neocortical-like processes advocates for understanding using simple learning principles. This discovery offers a ready avenue to improve the computational capacities of biologically plausible models.

### AUTHOR CONTRIBUTIONS

If you'd like to, you may include a section for author contributions as is done in many journals. This is optional and at the discretion of the authors.

### ACKNOWLEDGMENTS

Use unnumbered third level headings for the acknowledgments. All acknowledgments, including those to funding agencies, go at the end of the paper.

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
