# Supplementary Material for "Emergence of surprise and predictive signals from local contrastive learning"

## Abstract

This is the supplementary information for the "Emergence of surprise and predictive signals from local contrastive learning."

## A  Appendix A: equivalence between Forward-Forward architecture and Hebbian learning

In this section, we apply a simple argument to the underlying dynamics as result minimizing cumulative objective functions.

For a $N > 3$ Forward-Forward architecture where $N$ is the total number of layers, $I$ is the data input, and $\ell$ is the label input, the dynamics of each layer are governed by

$$\dot{\vec{x_1}} = \phi\left(W_1\hat{x}_1 + F_1 I(t') + B_1\hat{x}_2\right)$$
$$\dot{\vec{x_i}} = \phi\left(W_i\hat{x}_i + F_i\hat{x}_{i-1} + B_i\hat{x}_{i+1}\right)$$
$$\dot{\vec{x_N}} = \phi\left(W_N\hat{x}_N + F_N\hat{x}_{N-1} + B_N\ell(t')\right).$$

where $\hat{x}_i = \frac{\vec{x}_i}{|\vec{x}_i|}$ is the layer-normed pre-synaptic drive.

The locally defined loss for a one-step update takes the form of:

$$\mathcal{L}_{layer}(t') = (-1)^\eta \sigma(\vec{x}_i^T \vec{x}_i - T)$$

where $\sigma(x)$ represents the softplus as $\sigma(x) = log(1 + e^x)$ and is a smooth version of the ReLU nonlinearity. Importantly, the dynamics of the $\eta(t')$ are governed by bistable dynamics switching between:

$$\eta(t') = \delta_{L(t'),I(t')}$$

where $\delta_{ij}$ is the Kronecker delta notation. This bistable switching-like dynamics occurs with long timescales between switches and could have compelling correspondence with our understanding of neuromodulatory induced switching of the underlying dynamics. This also suggests a criteria for the selection of $\eta(t')$ on the instantaneous surprise of the stimulus against the speculative label. While beyond the scope of this work, the closure of this loop between activations and cost function may generate valuable insights into unsupervised variants of these learning rules.

We then execute a single step-gradient update in each parameter:

$$\partial_t W_i = -\alpha \nabla_{W_i} \mathcal{L}(t')$$
$$\partial_t B_i = -\alpha \nabla_{B_i} \mathcal{L}(t')$$
$$\partial_t F_i = -\alpha \nabla_{F_i} \mathcal{L}(t')$$

By iteratively minimizing this locally defined objective function, we seek a heiarhical structure which will work in concert with the other layers to minimize activations for positive data. However, when there is a data mismatch between label and image class, the representations will learn to avoid cancellation to maximize their surprise.

Indeed this single step update for a given layer takes the form of a Hebbian learning rule.

$$\nabla_{W_i}\mathcal{L}_i(t') =$$
$$(-1)^\eta \underbrace{\sigma'(\vec{x}_i^T(t')\vec{x}_i(t') - T)(\vec{x}_i(t'))}_{\text{Gating factor}} \underbrace{\phi'(W_i\hat{x}_i(t'-1) + F_i\hat{x}_{i-1}(t'-1) + B_i\hat{x}_{i+1}(t'-1))}_{\text{Pre-synaptic current}} \underbrace{\vec{x}_i(t'-1)}_{\text{post-synaptic firing rate}}$$

This takes the form a gated Hebbian or three-factor rule Bahroun et al.; Kuśmierz et al. (2017); Bredenberg et al. (2021); Pogodin & Latham (2020) linking the locally defined objective function to the product of the pre-synaptic current and the post-synaptic activation.Crucially, this loss for a single trial is indeed the cumulative sum over the full sequence:

$$\Delta x_i = \alpha \sum_{t'} \nabla_{x_i}\mathcal{L}_i(t')$$

These gradients have important implications on the shape of the learned solutions. These learned solutions (where the gradient goes toward zero can occur under a number of conditions). These conditions include the direct cancellation of the synaptic drive currents (input components) governing the time dynamics of hidden layer: $W_i x_i + F_i x_{i-1} + B_i x_{i+1} = 0$.

## A.1 LINEARIZING EVERYTHING AND REDUCING THE DIMENSION OF THE LAYERS TO ONE UNIT EACH

Our goal in this subsection is to stricly apply two simplifying assumptions to show how the inverted FF objective function enforces cancellation in an easily understandable environment. The first assumption is that the dynamics are linear. The second assumption is that each layer is only represented by a single unit to simply our view of cancellation. These assumptions enforce direct learned cancelation of top-down and bottom-up signals in these networks.

In the limit of linear dynamics of the underlying network and linear dynamics of the locally defined objective function:

$$\mathcal{L}_\rangle(t') = \sum x_i^2 - T$$

and

$$x_i = W_i\hat{x}_i + B_i\hat{x}_{i+1} + F_i\hat{x}_{i-1}$$

The gradients give rise to the simple learning dynamics of the form:

$$\dot{W}_i = \alpha(-1)^\eta x_i(t+1)x_i(t)$$
$$\dot{F}_i = \alpha(-1)^\eta x_i(t+1)x_{i-1}(t)$$
$$\dot{B}_i = \alpha(-1)^\eta x_i(t+1)x_{i+1}(t)$$

With the edge cases taking the form of:

$$\dot{B}_N\alpha(-1)^\eta x_i(t+1)L(t)$$
$$\dot{F}_1\alpha(-1)^\eta x_1(t+1)I(t)$$

These linear dynamics are determined by the discrete variable $\eta$ which tells you if the data is positive data or not. We can call then positive data where the labels are presented as classes and the images are presented as floats near the given class identity.

These nonlinear dynamics give us a rich learning sequence in which the we have fast time dynamics governing the layer population scalars $x_i$ and the slower dynamics of the learned parameters. This separation of timescales allows us to represent the mean population activity over the trial to study the convergence of the learning dynamics to steady state solutions.

The simulation of these linearized but still nonlinear dynamics for a three layer network in an online learning setting confirms our analytic for this one-dimensional projection of population activity in that $F \to -B$ over training timesteps 1.

For a one-layer architecture with similar top-down/bottom-up representations of image class and label respectively, the positive data equivalence trivially forces the feedforward $F$ and feedback $B$

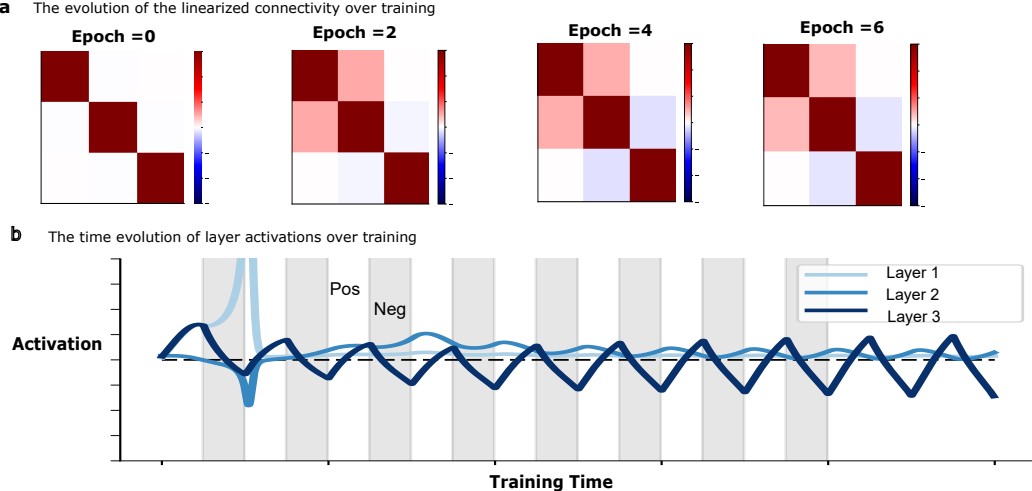

Figure 1: a) Over training the evolution of the simplified connectivity matrix develops opposing terms resulting in the cancellation of matched signals (top-down and bottom-up) into layer 2. b) The time course of training these linearized dynamics generates a system capable of switching between positive and negative data samples in an online fashion. Negative data is characterized by growing activations in layer 2 while positive data is characterized by cancelling activations in layer 2. These linearized dynamics give us a simplified playground to understand the emergence of cancellation with simple local learning rules.

matrices to converge toward the negative of each other, $B \rightarrow -F$. This can be seen in the evolution of the above dynamics of the weight vectors. The only way that the linearized dynamics can go to zero is when $\dot{B}_1 = 0 = -x_i L(t)$ and thus since $L(t)$ is fixed at a non-zero value by the supervision, the $x_i$ must go to zero. To achieve this, we must have clean cancellation of the underlying dynamics in the hidden layer. This forms the basis of the positive data cancellation and finds solutions which are consistent with increasing activation in response to mismatch or surprise.

## B   APPENDIX B: TEMPORAL DYNAMICS OF LAYER SIMILARITY

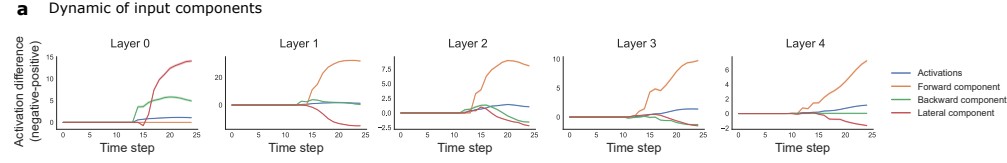

Figure 2: a) Layerwise input components differences (negative minus positive) dynamics across the timesteps. Noticeably the forward component is higher for negative data compared to positive data. We notice that for layer 0 the forward component is trivially identical for positive and negative data as it is driven by the input image.

We expanded the cosine similarity analysis across all timesteps for any two consecutive layers to further investigate this phenomenon. Figs. (3a) to (3b) illustrate the cosine similarity between activations of layers 2 and 3 across any two timesteps during the processing phase for both positive (Fig. (3a) left panel) and negative (Fig. (3b) left panel) data. For positive data, cosine similarities decrease over timesteps confirming that activations across different layers decorrelate over this period. On the other hand, for negative data, similarities increase over the same period, confirming and generalizing our findings in the main text. This analysis demonstrates the emergence of a striking temporal ordering of cancellations (positive data) and activations (negative data) which reflects the structurally imposed hierarchy of the layers. This relationship emphasizes the importance of a mechanistic understanding going beyond the naive cancellation of image and label representations as their first collision.

Figure 3: a) Cosine similarity analysis across all timesteps of layer 2 and 3 (left panel). Similarity Difference metric (SD) across timesteps between layer 2 and 3. b) Same as panel d but for negative data.

To examine the temporal dynamics further, we analyzed the difference between such similarities: for any pair of time steps, we computed the following metric. Denoted with $cos(a_{l2}(t_1), a_{l3}(t_2))$ is the cosine similarity between the activations $a_{l2}(t_1)$ of layer 2 at time $t_1$ and the activations $a_{l3}(t_2)$ of layer 3 at time $t_2$. We also defined the Similarity Difference $SD_{l23}(t_1, t_2) = cos(a_{l2}(t_1), a_{l3}(t_2)) - cos(a_{l2}(t_2), a_{l3}(t_1))$. This quantity provides insight into the temporal dynamics because, for $t_1 < t_2$, it is positive if the similarity between earlier activations in the first layer and later activations in the second layer is greater than the similarity between later activations in the first layer and earlier activations in the second layer. This value quantifies when current signals in one layer are analogous to subsequent signals in a second layer for any given timestep. A positive SD above the diagonal (accompanied by a negative SD below the diagonal) quantifies the influence of the first layer on subsequent timesteps in the second layer. This case, as described, is what we observed for negative data, confirming a bottom-up flow in late timesteps, Fig. (3b). For positive data, a top-down signal appears to flow into layer for a few time steps before activities across layers decorrelate and cancellation of activity occurs, Fig. (3a) right panel.

This analysis validates the presence of two information flows for positive and negative data, with distinct temporal relationships between layers. It further illustrates that such information flows have specific dynamical properties across layers, where the activity in a given layer precedes or follows the activity in others across the hierarchy, enabling the generation of predictive types of signals.

### B.1 Visualizing the image specific low-dimensional dynamics

A temporal analysis of the same latent space in two dimensions was conducted on the higher-order PCs with stronger class representation (Fig. (4a)). Activations start in the middle of the represented structure, before diverging and returning back to the beginning. This analysis shows behavior corroborating the above looping mechanics, driven by the recovery of the initial low activation state after initial excitement(Fig. (4b)).

### B.2 Label information and principal components

To understand if the principal component representation of the network dynamics effectively captured the variance of the underlying data, we trained a slew of multilayer perceptrons (MLPs) on the latents of 1000 samples at every timestep, reduced in dimensionality by a sliding-window of three PCs. This analysis shows high label decodability for the PCs plotted above which showed cleaner separability (4-6) (**??**), motivating the choice of these particular PCs. Additionally, most of the variance within this data is captured within the first 20 PCs, with decodability dropping to chance levels for PCs greater than 20.

## C  Appendix C: Predictive coding

In our predictive coding network (PCN), the inference and learning phases optimize the same following loss function:

$$\mathcal{L}(t) = \sum_i \|\vec{x}_{i-1} - \phi(B_i \vec{x}_i)\|_2^2,$$

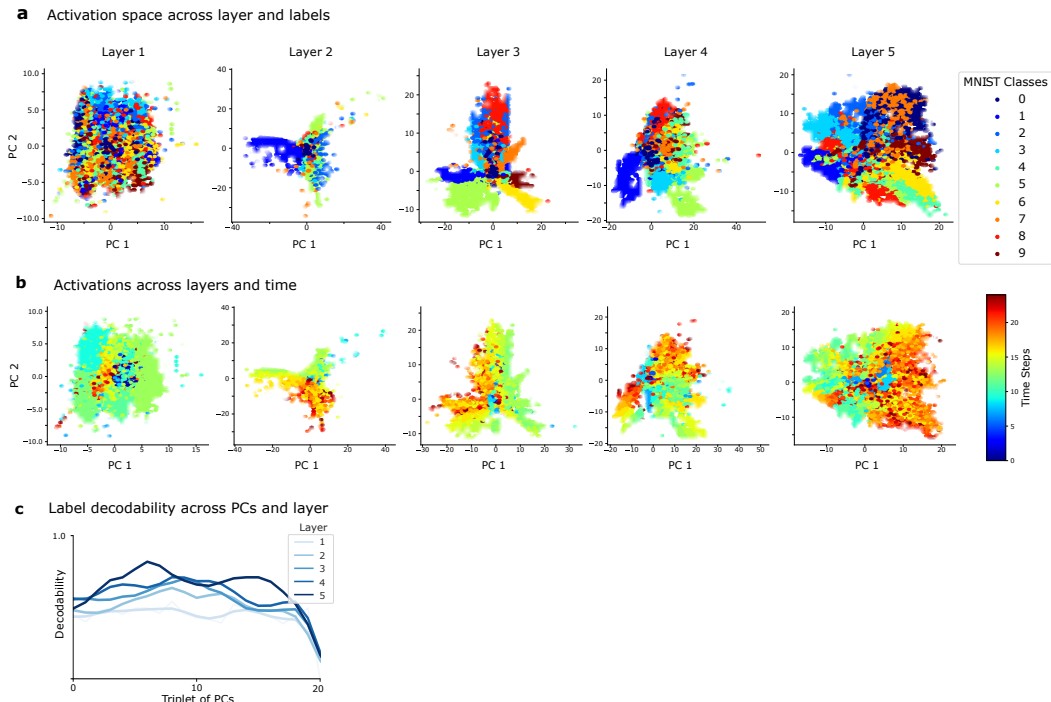

Figure 4: a) Representation of the layer-wise latent spaces on two dimensions via PCA where classes are represented by color. b) Same latent space representation, but color-coded based on timestep. c) Decodability (y-axis) across different layer's PCs (x-axis), indicating that PCs 4-10 capture rich representations. The x-axis label indicates the lower most PC out of the triplet used for decoding.

where $x_0$ is clamped to the input image.

The inference phase then takes the form of minimizing the loss with respect to the neural activities for each layer $i$:

$$\dot{\vec{x}}_i = \nabla_{\vec{x}_i}\mathcal{L}(t) = 2B_i^\top \frac{\mathrm{d}\phi^\top}{\mathrm{d}\vec{z}_i}\left(\vec{x}_{i-1} - \phi(B_i\vec{x}_i)\right) + \phi(B_{i+1}\vec{x}_{i+1}) - \vec{x}_i,$$

where $\vec{z}_i = B_i\vec{x}_i$. which is equivalent to a leaky neuron subject two sources of synaptic drive: 1) the feedback from top-down, and 2) the prediction error change. Allowing this to evolve to convergence gives us $\vec{x}_i^\star$.

The learning phase then adopts a gradient descent on the same loss with respect to $B_i$ for each layer $i$:

$$\dot{B}_i = \nabla_{B_i}\mathcal{L} = \frac{\mathrm{d}\phi^\top}{\mathrm{d}\vec{z}_i^\star}(\vec{x}_{i-1}^\star - \phi(B_i\vec{x}_i^\star))(\vec{x}_i^\star)^\top,$$

where we define $\vec{z}_i^\star = B\vec{x}_{i+1}^\star$ which is the convergence input current. Usefully, this one-step gradient also takes the form of a three-factor rule which combines pre-synaptic current, post-synaptic activity and a third gating or 'gain' factor. In this case, the third factor takes the form of the prediction error.

If we relax the convergence of assumption of the 'inference phase' and simply conduct online learning on this cost function, we can directly compare these updates to the inverted-FF model through comparison of our three-factor terms.

## C.1 COMPARISON BETWEEN INVERTED FF, PCN AND SUPERVISED UPDATE RULES

In the previous section we demonstrated that the update equations for the feedback weights $B$ evolve with a distinct third-factor responsible for gating the Hebbian updates of the weights in PCN. This

form of the rule is notably quite different from the inverted FF. To round out our comparison to include a simple variant of supervised loss we include the third factor for a supervised loss. We choose a simplified variant (with no feedback nonlinearities) of random feedback to focus our attention on the form of supervisory error Lillicrap et al. (2016).

These third factors for distinct learning rules can now be compared on the same standing:

$$\text{Third Factor} = \begin{cases} (-1)^{\eta} \sigma'(\vec{x}_i^T(t')\vec{x}_i(t') - T)(\vec{x}_i(t')) & \text{for the inverted FF} \\ (\phi(B_i \vec{x}_{i+1}) - \vec{x}_i) & \text{for the PCN} \\ \left( \prod_{j=N}^{i} B_j (\phi(\vec{x}_N) - y^{\star}) \right) & \text{for a supervised signal} \end{cases}$$

We emphasize two primary differences between the inverted FF and the PCN third factors.

The first is the presence of the contrastive sign flip designed to avoid the collapse of the dynamics onto the trivial solution. This constrastive term plays the role of the supervisory signal in which the information about the clamped label is passed to each layer through the top-down feedback and the global error signal (reminiscent of neuromodulator volume transmission) driving either the elimination or increae of surprise signal.

The second chief difference is that the inverted FF conditions weight updates on surprise being above a threshold while the PCN conditions weight update upon the activity prediction error. This follows from the supervised case where in both models, the surprise (inverted FF) and the prediction error (PCN) are acting like error signals in the network gating the Hebbian Pre-synaptic, post-synaptic concidence update rules.