# OpenReview forum: "Emergence of Surprise and Predictive Signals from Local Contrastive Learning"
_ICLR.cc/2024/Conference — Submitted to ICLR 2024_

### Official Review · Reviewer_D7tC · 2023-10-30

**Soundness:** 3 good
**Presentation:** 4 excellent
**Contribution:** 3 good
**Rating:** 8
**Confidence:** 4

**Summary:**

The authors propose a slight modification of the forward-forward algorithm, in which activity is suppressed for positive instances and increased for negative instances. While this is a slight modification, the primary contribution of this research is the demonstration of several emergent dynamics of the trained network, which are then related to the concept of predictive coding and may replicate several experimental neuroscience findings.

**Strengths:**

The paper is overall well organized and concise. Taking a well-publicized learning rule and showing a slight modification to relate to neuroscience theories should go a long way to keeping an open dialogue between ML and theoretical neuroscience. Additionally, nearly all of the questions I thought of asking, such as order and direction of surprise and expected signals, were addressed over the course of section 3, suggesting that the authors have put forethought into these analyses.Beyond a clarification described below, I have only two minor concerns with this paper. First, I wonder how interest it may be to ICLR attendees outside of the bio-inspired crowd. Some minimal discussion and possibly an additional abstract sentence to explain how predictive coding may be of broader ML interest could help alleviate this. Secondly, the loss function requires an explicit positive/negative cue (eta). Historically this sort of global signal has been contentious in computational models. It may be worth emphasizing that this error signal is singular, and how this may be much more easily achieved than a global vector valued error/target signal.

**Weaknesses:**

Beyond clarifications described below, I have only two minor concerns with this paper. First, I wonder how interest it may be to ICLR attendees outside of the bio-inspired crowd. Some minimal discussion and possibly an additional abstract sentence to explain how predictive coding may be of broader ML interest could help alleviate this. Secondly, the loss function requires an explicit positive/negative cue (eta). Historically this sort of global signal has been contentious in computational models. It may be worth emphasizing that this error signal is singular, and how this may be much more easily achieved than a global vector valued error/target signal.

**Questions:**

1.	In section 3.0 the optimizer is defined as RMSProp operating on the loss defined in equation 1. My understanding however is that standard RMSProp implementations calculate chained gradients and applied as a global optimization (though possibly with sub-losses defined on each layer), leading to weight transport and making the model non biologically plausible. If a modification, such as gradient stopping, was applied to prevent this global optimization then it should be described.
2.	Section 2, just before figure 2, states that only intralayer weights are trained, suggesting that recurrent (W)eights are, while biases (theta), (F)orward and (B)ackward weights are not. However, the surrounding text obfuscates this by speaking of the weights as “local properties”. If this is the case, that is a critical aspect of the model, and should highlighted and compared to a version in which the remaining parameters are trained. It would be beneficial to highlight in a single location which parameters do/don’t update, as well as defining abbreviations.
3.	If these questions are resolved/clarified, I would be happily convinced to increase my soundness and overall score.

---

> ### Author Response · Authors · 2023-11-23
> **Author Reply**
>
> Thank you for your constructive feedback on our paper. We are pleased that you found our paper well-organized and our approach relevant to the dialogue between machine learning and theoretical neuroscience. We have addressed your concerns and questions as follows:
>
> **Re: Supervisory signal limiting biological plausibility**
>
> The critique concerning the supervised nature of the contrastive learning in the model raises an important point about biological plausibility. While the model employs supervised learning, it is important to note that this approach is in the context of a field where the boundary between supervised and unsupervised learning mechanisms, and their roles in biological systems, is still an active area of research.
>
> The use of the singular error signal, η, in the loss function of the model can be seen as a reflection of a simplified global learning signal, which could, in some respects, be analogous to the role of neurotransmitters in biological neural networks. This analogy suggests that just as η provides a generalized feedback mechanism to the entire model, neurotransmitters in the brain can act over a wide area (via volume transmission) to modulate the activity of many neurons, rather than just those directly connected by synapses. This is a way to understand how global signals might occur in response to learning or other processes.
>
> We appreciate this feedback and have included the above details relating to the supervisory signal in our manuscript.
>
> **Re: Bioplausibility of RMSProp optimizer**
>
> Although the RMSProp optimizer utilizes moving averages of gradients, we don’t believe that observation in itself renders it biologically implausible. For example, it is biologically plausible for a neuron to use local information to adjust its learning. Classic rules like BCM theory leverage such history dependence to reduce the variance of the updates. Thirdly, we reset the parameter gradients each time a layer trains, ensuring the independence and localization of learning at each layer. Finally, all presynaptic input tensors are detached, acting as a stop grad, which is critical in preventing non-local weight updates. These modifications collectively serve to lend our implementation of Forward Forward as much as possible with local learning. We have updated the text to explain each of these points.
>
> **Re: Question regarding which weights are trained**
>
> We apologize for the confusion caused by our description. We mean that each layer is its own learning agent. When training is performed on a specific layer, only the weights connected to this layer are updated (i.e. forwards, backwards, lateral). We have updated this description to eliminate confusion.
>
> **Miscellaneous**
>
> We have also added some information to our abstract about the broader applicability of predictivity coding, and taken additional care in mentioning that this global error signal is simple and singular rather than vectored and complex.

---

### Official Review · Reviewer_fE2s · 2023-10-31

**Soundness:** 3 good
**Presentation:** 3 good
**Contribution:** 3 good
**Rating:** 8
**Confidence:** 4

**Summary:**

The paper presents a novel and biologically plausible mechanism for understanding cortical processing, focusing on predictive aspects. It introduces a model based on the Forward-Forward approach, a form of contrastive learning, and makes the following three contributions.


1) Hierarchical and Temporal Predictive Properties: The model reproduces some hierarchical and temporal aspects of predictive computations. It generates information flows that lead to surprise and cancellation signals, demonstrating its ability to capture, to some degree, the spatiotemporal predictive nature of cortical processing.

2) Mechanistic Understanding of Information Flow: The paper offers insights into the emergence of information flows by tracing their origins to the ability of the neural circuit to implement spatiotemporal cancellation across different layers of activity. This understanding could explain how the brain processes and predicts sensory information.


3) Biological Plausibility: The model's contrastive learning rule is shown to be equivalent to a unique form of three-factor Hebbian plasticity, which has strong connections to predictive coding. This highlights the biological plausibility of the proposed model. The training process involves positive and negative datasets, where surprise modulates layer activity based on whether the label and input match. The surprise calculation is defined in terms of individual layer activations. Importantly, training only involves forward passes, making the Forward-Forward algorithm biologically plausible and avoiding the non-locality of weight information inherent in backpropagation.

**Strengths:**

The paper is well written.

It provides substantial novelty in introducing a training procedure that mimics signal processing in biological networks, distinguishing between the presentation and processing phases. Surprise signals arise when there is a mismatch between sensory inputs and top-down signals (label information), similar to how cortical processing operates.

The authors pay significant attention to the interpretation of the learning and how this could be implemented in the brain. Obviously, it is to some degree biologically plausible, and the existing literature that looks into cortical microcircuits for predictive coding could be a good thing to add to discuss the possible implementation of the algorithm.

**Weaknesses:**

Although the paper offers a theoretically sound model, it lacks empirical validation through experiments or real-world data.

Including practical applications or experiments to support the model's claims would enhance its credibility and applicability in real-world scenarios.

**Questions:**

Could you elaborate more on the algorithm, as in presenting some pseudo-code as well as a description of the neural architecture used that can be more easily connected to the standard literature of artificial neural networks?

---

> ### Author Response · Authors · 2023-11-23
> **Author Reply**
>
> Thank you for your thoughtful and detailed review of our paper. We appreciate your positive feedback on the novelty and biological plausibility of our model, as well as your suggestions for improvement.
>
> We acknowledge your concern regarding the lack of empirical validation through experiments or real-world data. To address this, we have plans to extract from existing experimental data such as that done on the change detection task in mice[1]. This will not only provide empirical support for our model's claims but also demonstrate its applicability in real-world scenarios.
>
> In response to your question about a more detailed explanation of our algorithm, we have enhanced and consolidated the model architecture descriptions which were previously spread over multiple sections. Additionally, we have open sourced the code of our model, which is intended to aid in transparency regarding our methods.
>
> [1]Visual Behavior Neuropixels - brain-map.org. Retrived on 11/20/2023 from https://portal.brain-map.org/explore/circuits/visual-behavior-neuropixels.

---

### Official Review · Reviewer_3133 · 2023-11-01

**Soundness:** 2 fair
**Presentation:** 2 fair
**Contribution:** 2 fair
**Rating:** 3
**Confidence:** 4

**Summary:**

Backpropagation in deep networks has been widely acknowledged to suffer from a lack of biological plausibility.  Recently, a novel framework has been proposed to leverage contrastive learning on matching and non-matching datasets to learn only through forward passes through the network, avoiding several of the least biological features of backpropagation.  The authors utilize this approach to model neural correlates of surprise within a predictive learning framework, finding network dynamics that are broadly consistent with neuroscientific findings: stimuli that match “predicted” labels result in little activity, while those that do not result in large deviations from baseline activity.  They further analyze the resultant spatiotemporal dynamics using PCA, characterizing both the dynamics and decodability of the network across space (layers) and time. Finally, the authors observe that the trained networks result in a local 3-factor Hebbian rule, and perform an analysis on the linearized form to provide additional intuitive insight into the observed prediction-cancellation process.

**Strengths:**

The authors provide a strong biological background and motivation for the importance of moving beyond backprop-based methods to better understand learning in cortex, and more specifically for predictive learning that their results qualitatively match

**Weaknesses:**

_Major_

1. While I am unaware of other explicit work that matches the qualitative results the authors provide, the overall framework, including the revised loss function and connection to predictive coding (including the cancellation process demonstrated by the authors) were all explicitly anticipated in Hinton 2022.
   -  Moreover, the usage by the authors of a supervised contrastive learning process somewhat lessens the local learning process that can strengthen the biological plausibility of the results.  In particular, even though the learning process does not rely on propagating error information from the top layer, the local layer loss functions still contain explicit information about whether the inputs match the labels ($\eta(t)$). The novelty would be increased by adding a more biologically realistic unsupervised learning process to the network (e.g., SimCLR).
2. The PCA section does not seem to add much of substance to the manuscript
   - Different numbers of components are included in different analyses for reasons that are somewhat difficult to discern (see Questions)
   - Fig. 4a (top) seems to essentially describe the periodic patterns for the positive data observed in Fig. 2b.  Similarly for the periodic aspects in Fig. 4a (bottom) and Fig. 4c.
   - Fig. 4a (bottom) is used to illustrate the strong visual decodability of PCs 4-6 vs. 1-3 (top).  However, in Fig. 4d we observe that PCs 1-3 in fact allow for decodability seemingly well above chance (if the x-axis is at a decodability value of 0), and that this decodability does not substantially increase for layers 1 and 2 and perhaps even 3.
   - Fig. 4c does not seem to lead to much more insight than Fig. 4a (bottom), together with the observation that each loop in 4a (bottom) starts and ends in the center, as described for 4c.  This would be even more so if the trajectories in 4a (bottom) were modulated in intensity/luminosity/transparency to correspond to the associated time steps.
   - In Fig. 4e, the presentation phase ends before the associated decodability traces either stabilize or peak (see Questions)
   - Similarly, it seems as though the negative dynamics may cycle quasi-periodically as well based on Fig. 2b if enough time steps were included
   - Overall, it seems as though the full-network analysis approach as shown in Figs 2-3 comprise a better analytical approach, given the generally simple network dynamics
3. The model and training regimen are not explained well, and spread over several sections
   - The model description is terse in the Introduction, with further details added in Sec. 3.0 and 3.4.  However, the notation between the Introduction and Sec. 3.4 (and App. A) are inconsistent (x vs. r, $\theta$ vs T).  Similarly for training details
   - The readability would be improved by providing all of the model and training data in the Introduction, before any results are described

*Minor*
1. I could not find any reference to App. B in the main text
2. The final Results section (3.4) observes that the local aspect of the learning process is Hebbian.  While biologically important, this seems to fall directly out of the FF framework itself and only deserving of a passing observation
3. The linear dynamics results in App. A.1 allow for the input to approximately equal the label.  While such a representation could be learned, I don’t see how such a direct approximation is justified.
4. “Surprise” seems to be defined in two ways (“activity as surprise,” p. 2 top, then defined as components of the layer loss function and in terms of the activity on the bottom of p. 2 in Eq. 1)
5. Some typos (e.g., “at each time $x_{layer}(t’)$” p.2; “dynamics dynamics” in Supp. p. 2; “three-paired” p. 5; lack of equal signs in Supp. p. 2, $t$ vs. $t’$ in $z(t-1)$ on p. 7--also with and without arrow-vector decoration)
   - Similarly, some mathematical notation not explicitly described, though easily inferred ($L(t)$, $I(t)$, $\el(t)$)
   - While "PC" is implicitly defined by reference to PCA in Fig. 4's caption, both PCA and PC should initially be explicitly referenced with a full name (e.g., "Principle Component Analysis") in the main text before using the abbreviation
6. Fig. 3b would be more compelling if traces included those from several different network instantiations and included error shading
7. Figs 4d-e would provide some additional intuition if they were to being at a decodability of 0 and include a dashed horizontal line corresponding to chance levels of decodability for comparison purposes

**Questions:**

1. Fig 4d — what are the “Triplet of PCs” indicated in the abscissa label?  In the caption, it instead indicates they simply are the PCs, while in the main text it’s stated they are “sliding-window three-paired PCs.”   Please clarify what exactly is being decoded and provide consistent labels/descriptions in the manuscript
2. Why are the different principal components chosen in the analyses?  How many are included seems to confusingly vary from one analysis to the next with little or no explanation.
   - E.g., components 1-3 and then 4-6 are used in the primary analysis in Fig. 4a-c. Then the first 20 components are used for Fig 4d—while the main text indicates that “most of the variance is captured within the first 20 PCs,” it is not indicated either how much of the variance is captured, or if this is the reason for going to 20 but no further.  Graphically, there is a clear drop-off at 20, perhaps to within-chance levels of decodability. Perhaps this is the reason for choosing 20?
   - Finally, for Fig. 4e, 50 PCs are chosen.  Why are 50 chosen? Based off of Fig. 4d, it seems as though a more principled analysis would only include the first 19 or 20 or so components—ie, the ones that allow for above-chance levels of decodability.
3. How would the PCs in Fig. 4e continue to evolve if the presentation phase were prolonged?  In the figure, all PCs have an upward trajectory before the processing phase begins, making it difficult to discern how much of the continued rise during the processing phase is due to continued, prolonged and perhaps delayed dynamics arising during the presentation phase.  A more compelling analysis might be to allow the dynamics to either stabilize or peak before ending the presentation phase and beginning the processing phase.
4. Do the authors use layer normalization as per Hinton 2022?  Or is there a reason to not be concerned about the corollaries mentioned there in the authors’ model?

---

> ### Author Response · Authors · 2023-11-23
> **Author Reply**
>
> We are grateful for the thoroughness of your review and the opportunity to clarify and improve upon our work. We have addressed your comments and questions as follows:
>
> **Re: Forward-Forward framework novelty**
>
> We appreciate the reference to Hinton 2022 and acknowledge the foundational work it established. While Hinton anticipated an inverted loss function, our work is aimed at implementing this anticipated network in an attempt to model cortical activity, and exploring the emergent dynamics of spatiotemporal cancellation across layers. The enhancements our work provides in interpretability applied to this architecture will enable more effective future comparisons, taking the FF one step closer to neural dynamics.
>
> **Re: Supervisory signal limiting biological plausibility**
>
> The critique concerning the supervised nature of the contrastive learning in the model raises an important point about biological plausibility. While the model employs supervised learning, it is important to note that this approach is in the context of a field where the boundary between supervised and unsupervised learning mechanisms, and their roles in biological systems, is still an active area of research.
>
> The use of the singular error signal, η, in the loss function of the model can be seen as a reflection of a simplified global learning signal, which could, in some respects, be analogous to the role of neurotransmitters in biological neural networks. This analogy suggests that just as η provides a generalized feedback mechanism to the entire model, neurotransmitters in the brain can act over a wide area (via volume transmission) to modulate the activity of many neurons, rather than just those directly connected by synapses. This is a way to understand how global signals might occur in response to learning or other processes.
>
> We appreciate this feedback and have included the above details relating to the supervisory signal in our manuscript.
>
> **Re: PCA Analysis Clarity**
>
> We recognize the need for greater clarity in our PCA analysis and the rationale behind the number of components chosen for each figure. We have now revised these sections and figures to provide a clear explanation and the reasons for the chosen components. We have removed figures which do not serve to drive home our broader points.
>
> **Re: Description of the model**
>
> We appreciate the reviewer for noting the lack of clarity around model architecture. We have revised the structure of the paper to formally present the model at the beginning, and give a more clear explanation.
>
> **Re: Linear analysis in appendix A.1**
>
> Indeed, the linear dynamics description is confusing. In the original inverted FF we are not forcing the input approximately equal to the label. Rather, with respect to a layer, the forward (input-driven) representation and the backward (label-driven) representation are approximately equal. However, in the linear dynamics where we apply strong simplifications to drive intuition. Specifically, we enforce a single-neuron-per-layer assumption. Here, this transforms both the label and the input into a one-dimensional quantity which is convenient to consider as approximately equal during positive data and not equal during negative data for this three-layer, single-neuron linear dynamics.  We have updated the text to make this more clear and to emphasize the contribution of this simplified argument.
>
> **Miscellaneous**
>
> We thank the reviewer for their thoroughness in reviewing our article. The following contains our response to the minor issues and the answers to the questions raised.
> 1. Inconsistencies in notation have been corrected to ensure a coherent presentation throughout the manuscript. The following minor changes have been addressed in the text:
> 2. Added explicit references to Appendix B in the main text.
> 3. Clarified the definition of "surprise" by referring only to the level of activity as surprise.
> 4. Corrected typos and ensured consistent mathematical notation.
> 5. Defined PCA and PC explicitly before abbreviation.
>
>
> Regarding the explicit questions:
> - Question 1: Figure 4d: we have clarified the labels and descriptions regarding the "Triplet of PCs" and ensured consistency across the manuscript.
> - Questions 2 & 3: PCA Component Selection: The selection rationale is now articulated, including why certain numbers of components were included.
> - Questions 4: We do use layer normalization in this architecture, just as Hinton did. We have updated the text to explain this, along with the section 3.4.

---

### Official Review · Reviewer_mvwV · 2023-11-06

**Soundness:** 1 poor
**Presentation:** 1 poor
**Contribution:** 2 fair
**Rating:** 3
**Confidence:** 4

**Summary:**

This paper proposes a variation of the Forward-Forward Algorithm, which is an unsupervised learning model with local learning rules, using contrastive samples (positive and negative samples). The original Forward-Forward has as its objective maximizing the squared activity of each layer (greedily) for positive samples while minimizing it for negative samples. The variation proposed here uses the opposite objective, minimizing the activity of positive samples, and maximizing of negative samples, which makes it related to surprise signals and predictive coding models. The paper focuses on the dynamics of the model latent unit activity over time, inspecting for each layer, training phase and sample type. In particular, there is a difference between the dynamics of negative and positive samples, with cancellation happening in the bottom-up dynamics for positive samples. They also show the dynamics of the principal components of the latent space for different layers. Lastly, they derive the update rule from the proposed loss and map it to biologically plausible three-factor Hebbian learning rules.

**Strengths:**

The two main contributions of the paper:
- a potentially novel local surprise / predictive coding loss for contrastive samples in multi-layer networks;
- analysis of the dynamics of the latent variables for the hierarchical model.

Relating the loss function and activity of artificial deep networks to cortical activity is a promising approach for revealing how the cortex learns representations.

**Weaknesses:**

The main weaknesses of the paper are:
- lack of comparison with previous models, especially with predictive coding models;
- unclear insights brought by the analysis of the dynamics, especially concerning previous literature, alternative models and experimental findings;
- weak theoretical development.

While the authors motivate their model as a variation of the Forward-Forward model, since it is a predictive coding-like model, the authors should also relate it to previous prediction coding models (Golkar et al., Neurips 2022), biologically plausible models (Illing et al., Neurips 2021) and related literature (Grill et al., Neurpis 2020). How is this model similar or different from previous models? Is this a novel procedure? Does it work better or worse than other models?

Without further motivation and contextualization for the model, the in-depth dynamical analysis is also difficult to assess. How would the dynamics of similar models behave? In any case, if the model is novel, it should be motivated and analyzed further, before diving into such specific properties. As the authors motivate the model as a potential model for cortical learning and activity, there should be more precise references to data and papers on what these dynamics look like in the brain (e.g. Rabinovich, Huerta, Laurent, 2008), and why they matter.

The theoretical argument of mapping the update to a three-factor learning rule is not convincing, as almost any update for a local loss in a neural network will have such a general form. The model should also be presented formally at the beginning of the paper.

Minor:

What's the difference between r_i and x_i? And threshold \theta or T?

The accuracy for different number of layers should be included in the supplementary material, including the result for alternative models.

**Questions:**

How is this model different from previous models? Is this a novel procedure? Does it work better or worse than other models?

How would the dynamics of similar models behave?

What experimental evidence on cortical activity does this model relate to and why do they matter?

Why is this model particularly biologically plausible compared to related models?

What's the difference between r_i and x_i? And threshold \theta or T?

What is the performance of the model for different architectures, and in comparison to other models?

---

> ### Author Response · Authors · 2023-11-23
> **Author Reply**
>
> We are grateful for the thoroughness of your review and the opportunity to clarify and improve upon our work. We have addressed your comments and questions as follows:
>
> **Re: Lack of comparison with previous models**
>
> We acknowledge the importance of comparing our model to existing predictive coding and biologically plausible models.
>
> To this end, we have added a new section in the paper that provides a direct comparison with a newly implemented predictive coding model. This comparison outlines the similarities and differences, substantiating the novelty of our approach and its performance relative to these established models.
>
> **Re: Description of the model**
>
> We appreciate the reviewer for noting the lack of clarity around model architecture. We have revised the structure of the paper to formally present the model at the beginning, and give a more clear explanation.
>
> **Re: Theoretical argument for the three-factor learning rule.**
>
> We thank the reviewer for their helpful critique on our three-factor learning rule. We have since clarified the significance of this three-factor claim within the manuscript by 1) adding a discussion of how three-factors both unifies and differentiates learning rules and 2) a new discussion targeting the value of this unique third factor to both the ML and neuroscience communities.
>
> The key difference of the three-factor learning form of the inverted FF compared to previous learning rules (Gerstner 2018, Kuśmierz 2017) is the global supervisory signal (-1)^\eta term which tells the network whether it should maximize or minimize activations. In ML, this functional difference drives home the power of contrastive techniques and underscores recent (since our submission) innovations in merging contrastive-inspired methods with predictive coding using modern variants of BCM-like learning rules (Srinath-Halvagal, 2023). In neuroscience, this viewpoint suggests novel hypotheses about the varied role of volume transmitted neuromodulators such as dopamine (Gerstner, 2018), serotonin, acetylcholine, and norepinephrine in STDP learning.

---

### Author Response · Authors · 2023-11-23
**Summary Statement**

We thank each of their reviewers for their thoughtful and meticulous feedback to help improve this work. We are delighted that the reviewers felt that our story shared "strong biological background and motivation" and was "well organized and concise". We are thankful for the opportunity to address each of the reviewers comments in what follows.

In bulk, we have:
- Completed new numerical experiments on a Predictive Coding Network and incorporated observational and analytic comparisions with the inverted FF to underscore its utility.
expanded upon the simple theoretical treatment to emphasize why a three-factor description opens up useful comparisons and new normatively guided hypotheses about the role of neuromodulators in local learning rules
- Reconfigured description of the model and improved its analysis, including new figure panels in the main manuscript and additional supplementary figures
- Improved our clarity and exposition throughout the manuscript. We added  new acknowledgment and viewpoint on the global supervisory signal. We reprioritized the PCs analysis, moving redundant findings to the SI. We added a discussion of the role of RMSprop in our results and biocompatible interpretations.

We appreciate the reviewers contributions toward improving this work and thank them for their consideration.

---

### Meta-Review · Area_Chair_tpZi · 2023-12-06

**Metareview:**

This paper develops a contrastive method for learning based on the forward-forward algorithm (Hinton, 2022). In-line with predictive coding theories, the authors invert the forward-forward approach and minimize the activity of positive samples while maximizing the activity of negative samples, wherein "postitive" and "negative" samples are defined as those that produce the correct output and those that do not, respectively. The authors show that this leads to some of the standard properties of predictive coding algorithms, such as surprise and cancellation signals. The authors also show the connection to three-factor learning rules and suggest that this demonstrates the biological plausibility of the proposed algorithms.

This paper was a borderline case, with mixed reviews (two 3s and two 8s). The positive reviewers felt that the paper was well-written and demonstrated interesting phenomena that made it worthy of publication. The negative reviewers noted the lack of comparison to existing biologically plausible learning algorithms, lack of theoretical explanation for the algorithm, and heavy emphasis on analyses that provided few real new insights.

After discussion, the AC was largely convinced by the negative reviewers. Notably, though this paper proposes an interesting idea, it is very preliminary and there is no derivation to demonstrate that it should work well. Added to this, the empirical results are very weak, showing worse performance on relatively toy benchmarks, like MNIST, than that obtained by other local, biologically plausible learning rules over the last several years (see e.g. Scellier et al., 2017l, eq-prop; Sacremento et al. 2018, dendritic microcircuits; and Payeur et al., 2021, burstprop). There are some advantages to this algorithm, such as the lack of any need for additional algorithms to deal with weight alignment, but arguably, this paper does not demonstrate that the proposed approach will work well in general, either analytically or empirically. As well, in the AC's opinion, the demonstration of surprise and cancellation signals is not that insightful, given that the algorithm is essentially designed to try to cancel out expected activity. Given these considerations, a decision of 'reject' was reached.

**Justification For Why Not Higher Score:**

But, after looking over the reviews, the rebuttals, the discussion, and the paper itself, I was convinced that this paper does not meet the bar of existing work in the biologically plausible learning algorithm field.

**Justification For Why Not Lower Score:**

N/A

---

### Decision · Program_Chairs · 2024-01-16

Reject